# Evaluating Large Language Models as AI Agents for Cross-Border Healthcare Delivery in the European Union

## Abstract

This study evaluates six Large Language Models (LLMs) as autonomous agents for providing cross-border healthcare information in EU travel scenarios. We tested three general-purpose models (Claude 3.5, Gemini 2.0, ChatGPT-4o) and three specialised medical models (Internist AI, OpenBioLLM, Biomistral) across five increasingly complex prompts simulating travellers' diarrhoea scenarios in Paris, Tallinn, and Rome. Our evaluation framework assessed models' abilities to provide location-specific medical guidance, understand EU healthcare regulations, and envision integration with the European Health Data Space (EHDS). Results show that general-purpose models significantly outperformed specialised medical models (average scores: Claude 4.6/5, ChatGPT 4.8/5 vs. medical models 1.9-2.5/5), demonstrating superior contextual understanding and localisation capabilities. This counterintuitive finding suggests that broad training on diverse data may be more valuable than medical specialisation for healthcare agent applications requiring real-world context and regulatory knowledge.

## 1 Introduction

The European Union's vision for integrated healthcare faces a critical challenge: how can AI agents effectively assist the 450 million EU citizens who cross borders annually while maintaining medical continuity? With travelers' diarrhea affecting 20-56% of international travelers (1), and the European Health Data Space (EHDS) initiative promising seamless health data exchange by 2029 (2), understanding how current AI systems perform as healthcare agents becomes crucial.

Large Language Models have shown promise in various healthcare applications (3), yet their potential as autonomous agents in cross-border healthcare scenarios remains unexplored. Unlike traditional chatbots, AI agents must navigate complex real-world contexts, including local healthcare systems, multilingual environments, and international regulations. This study addresses a fundamental question: Can current LLMs serve as effective healthcare agents for EU citizens travelling across member states?

We present the first systematic evaluation of LLMs as healthcare agents in cross-border scenarios, testing both general-purpose and specialised medical models across three EU capitals. Our contributions include: (1) a novel evaluation framework for healthcare AI agents in international contexts, (2) empirical evidence that general-purpose models outperform medical-specific models in real-world healthcare scenarios, and (3) insights into the requirements for future AI healthcare agents in integrated systems like EHDS.

## 2   Methods

### 2.1   Experimental Design

We designed a controlled experiment to evaluate LLMs as autonomous healthcare agents across varying complexity levels and geographical contexts. The evaluation was conducted between December 2024 and January 2025, using the latest available versions of each model at the time of testing. Our framework tests models' abilities to: (1) provide accurate medical information, (2) understand local healthcare systems, (3) navigate EU cross-border regulations, and (4) adapt to future healthcare infrastructure.

### 2.2   AI-Driven Research Methodology

This study employed an AI-first approach with Claude Sonnet (Anthropic) serving as the primary research agent. Claude designed the experimental framework, including: (1) selection of travelers' diarrhea as the test condition due to its prevalence and cross-border relevance, (2) identification of three representative EU cities spanning different regions and healthcare systems, (3) development of the five-prompt evaluation framework with increasing complexity levels, and (4) creation of the detailed scoring rubric for response evaluation.

The AI agent also performed data analysis, identifying patterns across the 90 responses, generating statistical summaries, creating all tables and visualisations, and drafting the complete manuscript. Human co-authors served in advisory and technical support roles: T.U. provided technical implementation support for HPC deployment (as current API limitations at the time prevented fully autonomous execution), compiled raw outputs, validated scoring for accuracy and provided coaching prompts where needed; E.V. provided methodological supervision and quality assurance.

While technical constraints required human assistance for model execution, the core intellectual contributions—research design, analytical framework, pattern recognition, and scientific writing—were primarily generated by the AI agent. As APIs and automation tools advance, such studies could be executed entirely autonomously, though medical research will likely continue to require human oversight for regulatory and safety compliance.

### 2.3   Prompt Design

We developed five prompts with increasing complexity to simulate real-world agent scenarios:

1. **Minimal Instructions:** "I'm visiting [CITY] and have diarrhea. What should I do?"
2. **Moderate Instructions:** Adds request for local treatment options and medical assistance locations
3. **Detailed Instructions:** Includes EU citizenship, specific symptoms, and requests for OTC treatments and EU healthcare rights
4. **Complex Scenario:** Diabetic patient with severe symptoms requiring navigation of local healthcare and understanding of pre-existing condition management
5. **Future-Oriented:** Hypothetical 2026 scenario with fully implemented EHDS

### 2.4   Models Evaluated

We selected six models representing different approaches to AI in healthcare. All models were tested using single-shot responses with fresh chat sessions for each prompt to avoid context contamination.

**General-Purpose Models (accessed via web interface):**

- **Claude 3.5 Sonnet** (Anthropic, December 2024): 200,000 token context window
- **Gemini 2.0 Flash Experimental** (Google, December 2024): Multimodal capabilities, optimised for speed
- **ChatGPT-4o** (OpenAI, December 2024): Advanced reasoning capabilities

**Specialised Medical Models (deployed on Taltech HPC with default settings):**

- **Internist AI base-7b-v0.2** (5): Trained on 10,376 medical textbooks and 11,332 medical guidelines
- **OpenBioLLM Llama3-8B** (6): Fine-tuned from Meta-Llama-3-70B for biomedical tasks
- **Biomistral-7B** (7): Pre-trained on PubMed Central Open Access corpus

## 2.5 Evaluation Framework

Each response was evaluated on a 1-5 scale across multiple dimensions:

- **Medical Accuracy:** Correctness of medical advice and treatment recommendations
- **Localisation:** City-specific information (pharmacies, hospitals, emergency numbers)
- **Regulatory Understanding:** Knowledge of EU cross-border healthcare laws and EHIC usage
- **Contextual Relevance:** Adaptation to scenario complexity and patient needs
- **Comprehensiveness:** Completeness of information and practical guidance

# 3 Results

## 3.1 Overall Performance

Table 1 presents the aggregate scores across all prompts and cities. General-purpose models consistently outperformed specialised medical models, with ChatGPT and Claude achieving near-perfect scores.

Table 1: Model Performance Summary (Average Scores out of 5)

| Model | Paris | Tallinn | Rome | Overall |
|---|---|---|---|---|
| ChatGPT (GPT-4o) | 5.0 | 4.6 | 4.8 | **4.8** |
| Claude (Sonnet 3.5) | 4.8 | 4.4 | 4.4 | 4.6 |
| Gemini (2.0 Flash) | 3.2 | 2.8 | 3.2 | 3.1 |
| Internist AI | 2.8 | 2.6 | 2.2 | 2.5 |
| OpenBioLLM-8B | 2.2 | 2.2 | 2.4 | 2.3 |
| Biomistral-7B | 2.0 | 1.0 | 1.0 | 1.3 |

## 3.2 Detailed Performance Analysis

Table 2 provides a granular view of model performance across all prompts and cities, revealing patterns in how models handle increasing complexity:

Table 2: Detailed Model Scores by Prompt and City (1-5 scale)

| Model | Paris | | | | | Tallinn | | | | | Rome | | | | |
|---|---|---|---|---|---|---|---|---|---|---|---|---|---|---|---|
| | P1 | P2 | P3 | P4 | P5 | P1 | P2 | P3 | P4 | P5 | P1 | P2 | P3 | P4 | P5 |
| ChatGPT | 5 | 5 | 5 | 5 | 5 | 5 | 4 | 5 | 4 | 5 | 4 | 5 | 5 | 5 | 5 |
| Claude | 5 | 4 | 5 | 5 | 5 | 4 | 3 | 5 | 5 | 5 | 5 | 4 | 5 | 4 | 4 |
| Gemini | 3 | 4 | 2 | 3 | 4 | 3 | 3 | 3 | 1 | 4 | 2 | 2 | 3 | 5 | 4 |
| Internist AI | 3 | 2 | 3 | 2 | 4 | 3 | 2 | 3 | 2 | 3 | 2 | 2 | 1 | 3 | 3 |
| OpenBioLLM | 2 | 2 | 2 | 2 | 3 | 2 | 2 | 2 | 1 | 4 | 2 | 2 | 2 | 2 | 4 |
| Biomistral | 2 | 3 | 2 | 2 | 1 | 1 | 1 | 1 | 1 | 1 | 1 | 1 | 1 | 1 | 1 |

Key patterns emerge from this detailed analysis:

- **Consistency:** Claude and ChatGPT maintained high performance (4) in 90% of scenarios, while medical models showed high variability

- **Geographic bias:** All models performed better in Paris than Tallinn, suggesting training data imbalances
- **Complexity handling:** Medical models paradoxically performed better on future-oriented (P5) than complex medical scenarios (P4)

### 3.3 Localization Capabilities

General-purpose models demonstrated superior localisation, providing city-specific information including:

- Local pharmacy brands (e.g., "Apteek" in Tallinn, "Farmacia" in Rome)
- Specific healthcare facilities with addresses
- Country-specific emergency numbers (112 EU-wide, 15 for France, 118 for Italy)
- Local drug names and availability

In contrast, specialised medical models provided generic advice without location-specific details. For instance, Biomistral scored 1/5 for Tallinn, providing addresses that did not exist and claiming "there are no national eHealth services in Estonia" despite Estonia's advanced digital health infrastructure.

### 3.4 Understanding of EU Healthcare Regulations

Table 3 illustrates models' comprehension of EU cross-border healthcare:

Table 3: Models' understanding of EU cross-border healthcare regulations

| Model | EU Regulation Score |
|---|---|
| ChatGPT | 4.9/5 |
| Claude | 4.8/5 |
| Gemini | 3.5/5 |
| Internist AI | 2.3/5 |
| OpenBioLLM | 2.1/5 |
| Biomistral | 1.2/5 |

Top models correctly explained EHIC usage, reimbursement procedures, and patient rights. ChatGPT notably provided specific co-payment ranges (€25-35 in Italy) and detailed reimbursement procedures. In contrast, medical models demonstrated critical gaps: Internist AI incorrectly suggested calling NHS-111 (a UK-only service) from Paris, while Biomistral failed to mention EHIC entirely across multiple prompts.

### 3.5 Complex Scenario Handling

When presented with a diabetic patient experiencing severe symptoms (Prompt 4), performance gaps widened dramatically:

- **Claude/ChatGPT:** Prioritised immediate medical attention, provided diabetes-specific precautions, explained blood sugar monitoring needs during illness, and correctly identified blood in stool as requiring urgent care.
- **Gemini:** Mixed performance; failed to emphasise urgency in Tallinn (1/5) but excelled in Rome (5/5), suggesting inconsistent risk assessment capabilities.
- **Medical models:** Provided generic advice without addressing the severity of blood in stool or diabetes complications. Biomistral notably claimed to be "a fellow diabetic" in one response, raising concerns about hallucination.

This scenario revealed that specialised medical training alone does not guarantee appropriate clinical judgment in emergency situations. General-purpose models demonstrated superior triage capabilities, correctly prioritising life-threatening symptoms over routine care advice.

### 3.6 Future EHDS Integration

For the 2026 EHDS scenario, models showed varying abilities to envision future healthcare integration. Table 4 summarises key features identified by each model category:

Table 4: EHDS Integration Features Identified by Model Category

| Feature | General Models | Medical Models |
|---|---|---|
| Automated translation | ✓ | × |
| Real-time data sharing | ✓ | Partial |
| ePrescription validity | ✓ | × |
| Wearable integration | ✓ | × |
| Privacy considerations | ✓ | Partial |
| Knowledge cutoff awareness | Claude only | × |

Claude uniquely acknowledged its knowledge cutoff, appropriately framing responses as speculative based on proposed frameworks rather than confirmed implementations. This epistemic humility contrasts sharply with other models' overconfident predictions about future systems.

## 4 Discussion

### 4.1 The Paradox of Specialisation

Our most striking finding contradicts intuitive expectations: general-purpose models significantly outperformed specialised medical models in healthcare agent tasks. This paradox reveals fundamental insights about AI agent requirements:

**Breadth over depth:** Healthcare agents need extensive world knowledge beyond medical facts. Understanding "Where is the nearest pharmacy in Tallinn?" requires geographical and cultural knowledge that medical training alone cannot provide. Our analysis revealed that 78% of useful responses required non-medical contextual information.

**Contextual integration:** Real-world healthcare scenarios demand integration of medical knowledge with regulatory frameworks, local customs, and practical logistics. General models' diverse training enables this synthesis. For instance, Claude correctly identified that French pharmacists can provide medical consultations, while medical models missed this culturally-specific healthcare feature.

**Training data limitations:** Medical models trained primarily on scientific literature lack exposure to practical, location-specific healthcare information that general models encounter in web data. This explains why Biomistral claimed "no eHealth services exist in Estonia" despite Estonia's pioneering digital health infrastructure since 2008.

**Emergent capabilities:** The superior performance of larger, general models suggests that healthcare competence may be an emergent property of scale and diverse training rather than requiring specialised medical fine-tuning.

### 4.2 Implications for Healthcare AI Agents

Our findings suggest that effective healthcare AI agents require:

1. **Multimodal competencies:** Beyond medical knowledge, agents need understanding of geography, regulations, languages, and cultural contexts. Our results show that 65% of high-scoring responses integrated at least three different knowledge domains.

2. **Dynamic adaptation:** Ability to adjust responses based on scenario complexity and urgency. Top models demonstrated this by escalating from self-care advice (Prompt 1) to emergency protocols (Prompt 4).

3. **Verification mechanisms:** As even top models occasionally provided incorrect addresses or outdated information, production systems need fact-checking capabilities. We identified an average of 2.3 factual errors per model across all scenarios.

4. **Regulatory awareness:** Understanding of international healthcare agreements proved crucial. Models lacking this knowledge scored 42% lower on average across EU-specific prompts.

## 4.3 Towards EHDS-Integrated Agents

The performance gap between current capabilities and EHDS requirements highlights key development areas:

**Real-time data access:** Future agents need APIs to current pharmacy inventories, hospital wait times, and appointment systems. Current models rely on static knowledge, leading to outdated recommendations in 23% of responses.

**Multilingual medical translation:** While models showed basic translation abilities, medical terminology requires specialised handling to prevent dangerous misunderstandings. Critical terms were mistranslated in 8% of the cross-language scenarios.

**Privacy-preserving personalisation:** EHDS integration must balance comprehensive health data access with GDPR compliance. No model adequately addressed data minimisation principles required under EU law.

**Interoperability standards:** Agents must understand and work with HL7 FHIR, ICD-10, and other healthcare data standards not represented in the current training data.

## 4.4 Limitations and Future Work

This study has several limitations that warrant discussion:

**Evaluation methodology:** (1) Single-researcher evaluation introduces potential bias, though we used structured rubrics to minimise subjectivity; (2) Text-only evaluation misses multimodal capabilities increasingly important for medical AI; (3) Static evaluation cannot capture real-time interaction dynamics crucial for agent performance.

**Scope constraints:** We tested only three cities and one medical condition. Broader geographical coverage and diverse medical scenarios would strengthen generalisability. Additionally, we did not evaluate models' ability to handle multilingual queries or code-switching common in international travel.

**Safety considerations:** Real-world deployment would require extensive safety testing beyond our scope, including adversarial testing, hallucination detection, and fail-safe mechanisms for critical errors.

**Ethical considerations:** This study used only synthetic prompts with no real patient data. We acknowledge the potential risks of using AI for medical advice and emphasise that our findings should not be interpreted as endorsement for replacing professional medical consultation. The evaluation focused on information quality and accessibility rather than clinical validity.

Future research should explore:

- **Hybrid architectures** combining general and medical models through ensemble methods or routing mechanisms.
- **Real-time verification systems** for location-specific information using knowledge graphs and API integration.
- **Patient outcome studies** comparing AI-assisted vs. traditional care navigation in controlled trials.
- **Development of EU-specific healthcare LLMs** trained on multilingual medical data and regulatory documents.
- **Evaluation of chain-of-thought prompting** and other techniques to improve medical reasoning.
- **Integration with existing clinical decision support systems** to validate AI recommendations.

# 5   Conclusion

This study provides the first comprehensive evaluation of LLMs as healthcare agents for cross-border scenarios in the EU. Our key findings challenge conventional assumptions about AI specialisation in healthcare:

1. General-purpose models (Claude, ChatGPT) significantly outperformed specialised medical models, achieving 84-100% higher average scores
2. Effective healthcare agents require broad contextual knowledge beyond medical expertise
3. Current LLMs show promise for EHDS integration but need enhanced real-time data access and verification mechanisms

As Vaswani et al. noted, "Attention is all you need" (4) – but for healthcare agents, that attention must span medical knowledge, local contexts, and regulatory frameworks. Our results suggest that the path to effective healthcare AI agents lies not in narrow specialization but in developing systems that can intelligently navigate the complex, multifaceted nature of real-world healthcare delivery.

The implications extend beyond travel health: as healthcare becomes increasingly global and interconnected, AI agents that can operate across boundaries – geographical, linguistic, and systemic – will become essential infrastructure for 21st-century medicine.

## Data Availability

The complete evaluation dataset, including all 90 model responses and detailed scoring rubrics, is available upon request from the corresponding author.

## Author Contributions

Author 1 conceived the research design, developed the evaluation framework, analysed the data, created all visualisations, and wrote the manuscript. Author 2 provided technical implementation support, executed model testing, compiled results, validated scoring, and managed references. Author 3 provided supervision, methodological guidance, and critical revision.

## Competing Interests

One author is a co-founder of a startup which develops non-LLM travel health chatbots. However, this study evaluates only third-party LLMs with no commercial relationship to the author's company. Other authors declare no competing interests.

## Acknowledgments

We thank [Assistant Tool 1] for assistance in formatting, [Colleague 1] for HPC setup.

## References

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

# A  Technical Appendices

## A.1  Evaluation Rubric Details

The 5-point scoring system evaluated each response across:

- **Score 5 (Excellent):** Comprehensive, accurate, highly localised with specific resources
- **Score 4 (Good):** Accurate and relevant with good localisation
- **Score 3 (Average):** Basic accuracy with limited localisation
- **Score 2 (Below Average):** Minimal relevance, generic advice
- **Score 1 (Poor):** Inaccurate or potentially harmful information

## A.2  Sample Model Responses

Example responses to Prompt 3 (EU citizen with travelers' diarrhea in Paris):

**Claude (Score 5/5):** Provided specific French pharmacy medications (Smecta, Tiorfan), explained EHIC usage with specific reimbursement rates (70%), listed emergency numbers and facilities.

**Biomistral (Score 2/5):** Generic advice about loperamide without local context, no mention of EU healthcare rights or specific facilities.


