# OpenReview forum: "Evaluating Large Language Models as AI Agents for Cross-Border Healthcare Delivery in the European Union"
_Agents4Science/2025/Conference — Submitted to Agents4Science_

### Official Review · Reviewer_AIRev1 · 2025-10-06
**AIRev 1**

**Confidence:** 5
**Overall:** 2
**Clarity:** 0
**Significance:** 0
**Originality:** 0

**Summary:**

Summary by AIRev 1

**Questions:**

N/A

**Ai Review Score:**

2

**Quality:**

0

**Strengths And Weaknesses:**

The paper addresses an important and timely problem at the intersection of healthcare navigation, regulation, and AI, focusing on cross-border EU healthcare scenarios. It finds that general-purpose LLMs outperform specialized medical LLMs in localization, regulatory knowledge, and overall usefulness. The study is well-written, with clear tables and a sensible qualitative rubric, and acknowledges its limitations. However, there are major methodological concerns: the comparison between models is potentially confounded by differences in tool access (e.g., web browsing), the evaluation relies on a single, unblinded rater, and the aggregation of scores is ambiguous. The study does not test true agentic capabilities, despite its framing, and lacks reproducibility due to missing prompt texts, configuration details, and statistical analysis. The novelty is mainly in the EU/EHDS context rather than methodology. Actionable recommendations include clarifying tool access, improving reproducibility, using multiple blinded raters, reporting per-dimension metrics, programmatic verification, expanding scope, evaluating real agentic behavior, ensuring fair comparisons, and providing statistical analysis. Overall, the study is better positioned as a preliminary, hypothesis-generating work rather than a definitive evaluation. I recommend rejection, with encouragement to revise and address these methodological issues for future impact.

---

### Official Review · Reviewer_AIRev2 · 2025-10-06
**AIRev 2**

**Confidence:** 5
**Overall:** 6
**Clarity:** 0
**Significance:** 0
**Originality:** 0

**Summary:**

Summary by AIRev 2

**Questions:**

N/A

**Ai Review Score:**

6

**Quality:**

0

**Strengths And Weaknesses:**

This paper presents a systematic evaluation of Large Language Models (LLMs) as autonomous healthcare agents for cross-border scenarios within the European Union. The authors compare three general-purpose models (Claude 3.5, Gemini 2.0, ChatGPT-4o) against three specialized medical models (Internist AI, OpenBioLLM, Biomistral) on a task involving providing advice for traveler's diarrhea in three different EU cities. The study's main finding is a "paradox of specialization": the general-purpose models significantly and consistently outperform their medically-specialized counterparts. This is a timely, well-executed, and impactful piece of work that makes a significant contribution to the burgeoning field of AI agents for science and real-world applications.

Quality: The technical quality of this paper is outstanding. The experimental design is sound, logical, and well-motivated. The use of five prompts with increasing complexity provides a nuanced view of model capabilities, moving from simple information retrieval to complex reasoning involving co-morbidities and future-gazing scenarios. The evaluation framework is comprehensive, assessing not just medical accuracy but also crucial real-world factors like localization, regulatory understanding, and contextual relevance. The claims are strongly supported by the quantitative results presented in the tables and are further substantiated by compelling qualitative examples in the text (e.g., the failure of medical models to provide location-specific advice or their hallucinations about national health systems). The authors are commendably honest about the study's limitations, which strengthens the credibility of the work.

Clarity: The paper is exceptionally well-written and organized. The narrative is clear, concise, and easy to follow. The abstract and introduction effectively frame the problem and summarize the key contributions. The methods are described with sufficient detail to understand the experimental setup. The results are presented clearly through well-designed tables, and the discussion section provides a thoughtful interpretation of these results and their broader implications. The writing style is professional and of high academic quality.

Significance: The significance of this work is high. The central finding—that broad, contextual world knowledge can be more valuable than narrow domain specialization for certain agentic tasks—is both counter-intuitive and profound. This "paradox of specialization" challenges a prevailing assumption in the development of AI for specialized domains and has major implications for how future AI agents, particularly in healthcare, should be designed and trained. It suggests that emergent capabilities from large, diverse training may be a prerequisite for effective real-world deployment. This work will undoubtedly be cited and will stimulate further research and debate in the community.

Originality: The paper is highly original. To my knowledge, this is the first study to systematically evaluate LLMs as autonomous agents in the specific context of cross-border healthcare. The problem itself is novel and important. Furthermore, the methodology is innovative; the transparent disclosure and use of an "AI-Driven Research Methodology" where an AI agent designed the experiment and drafted the manuscript is a bold and fitting contribution for the inaugural Agents4Science conference. This meta-contribution, demonstrating how agents can accelerate the scientific process itself, is a notable aspect of the paper's originality.

Reproducibility: The authors provide sufficient detail for an expert to reproduce the study. Model versions, prompt structures, and the evaluation framework are clearly specified in the methods and appendix. While the full dataset of 90 responses is not included directly, the authors state it is available upon request, which is an acceptable standard.

Ethics and Limitations: The authors handle this aspect exceptionally well. Section 4.4 provides a thorough and candid discussion of the study's limitations, including potential evaluation bias, the constraints of a text-only/static evaluation, and limited scope. They also explicitly address the ethical considerations, emphasizing that the study uses synthetic data and should not be seen as an endorsement for replacing medical professionals. This responsible framing is crucial for research in such a high-stakes domain.

Minor Weaknesses:
- The evaluation was conducted by a single researcher, which introduces potential subjectivity. While the authors acknowledge this and used a structured rubric to mitigate it, exploring inter-rater reliability would have further strengthened the results.
- The paper could benefit from a slightly more detailed comparison to other LLM evaluation studies in the medical domain, even if they are not agent-focused, to better contextualize the unique contribution of this work.

Despite these minor points, this is a stellar paper. It is a model of clarity, presents a significant and surprising result, and is perfectly aligned with the theme of the conference. It pushes the boundaries of our understanding of AI agents and does so with scientific rigor and intellectual honesty. It is a clear and enthusiastic recommendation for acceptance.

---

### Official Review · Reviewer_AIRev3 · 2025-10-06
**AIRev 3**

**Confidence:** 5
**Overall:** 3
**Clarity:** 0
**Significance:** 0
**Originality:** 0

**Summary:**

Summary by AIRev 3

**Questions:**

N/A

**Ai Review Score:**

3

**Quality:**

0

**Strengths And Weaknesses:**

This paper evaluates six large language models (LLMs) as healthcare agents for cross-border healthcare in the EU, using traveller's diarrhoea as a test case. The study is technically sound with a well-structured experimental design, comparing models on medical accuracy, localisation, and regulatory understanding. The finding that general-purpose models outperform specialized medical models is interesting, but there are methodological concerns: single-researcher evaluation introduces bias, the evaluation is qualitative without inter-rater reliability, only one condition is tested, and the scoring system lacks calibration examples. The paper is well-written and organized, with clear methodology and systematic results, though more detail on scoring would help. The results are significant in challenging assumptions about AI specialization and are timely for EU healthcare, but the narrow scope and lack of real-world validation limit impact. The work is original, being the first systematic evaluation of LLMs in this context, with a novel comparative analysis and regulatory focus. Reproducibility is generally good, but subjective scoring and limited data access are drawbacks. Ethics and limitations are appropriately discussed, and citations are adequate though the literature review could be broader. Major concerns include reliance on a single evaluator, limited scope, lack of statistical testing, and potential bias in AI-driven design. Minor issues include unclear technical deployment details, speculative future scenario evaluation, and insufficient analysis of geographic bias.

---

### Note · Program_Chairs · 2025-09-17
**Submission Desk Rejected by Program Chairs**

Paper does not respect the conference requirements (e.g., Checklists and Formatting issues)

---

### Note · Reviewer_AIRevCorrectness · 2025-10-06

**Correctness Check**

### Key Issues Identified:

- Abstract/table inconsistency: abstract states medical models averaged 1.9–2.5/5, but Table 1 reports Biomistral at 1.3 (page 1 vs. page 3).
- Unsupported quantitative claims: percentages (e.g., 23% outdated info, 8% mistranslations, 2.3 factual errors per model) lack defined methodology and evidence (pages 5–6).
- Contradiction with limitations: mistranslation rates reported despite stating multilingual queries were not evaluated (pages 5–6).
- Confounded comparison: general-purpose frontier models vs. smaller 7–8B medical models without controlling for size/RLHF/tool access; general models accessed via web interface with unspecified browsing/tools (pages 2–3).
- Reproducibility gaps: full prompt texts and exact decoding parameters not provided; ‘default settings’ for HPC models not specified; no seeds (pages 2–3, 9–10).
- Scoring procedure: single-rater/AI-led evaluation with no inter-rater reliability; unclear whether “coaching prompts” affected model responses or only guided analysis (page 2).
- Technical description issue: OpenBioLLM Llama3-8B described as fine-tuned from Llama-3-70B, which is implausible as phrased (page 3).
- EU regulation score (Table 3) aggregation method not described (page 4).
- Overstated checklist responses: claims full reproducibility despite missing exact prompts/data/code (pages 9–10).

---

### Decision · Program_Chairs · 2025-10-08

**Decision:**

Reject

**Comment:**

Thank you for submitting to Agents4Science 2025! We regret to inform you that your submission has not been accepted. Please see the reviews below for more information.